# Detection and Tracking Method of Maritime Moving Targets Based on Geosynchronous Orbit Satellite Optical Images

**Fengqi Xiao**, **Fei Yuan** and **En Cheng** *

Key Laboratory of Underwater Acoustic Communication and Marine Information Technology,
Ministry of Education, Xiamen University, Xiamen 361005, China; uacxiaofengqi@stu.xmu.edu.cn (F.X.);
yuanfei@xmu.edu.cn (F.Y.)
* Correspondence: chengen@xmu.edu.cn; Tel.: +86-139-5016-5480

**Abstract:** The GF-4 geosynchronous orbit satellite can observe a large area for a long time, but the unique characteristics of its optical remote sensing image restrict the detection of maritime targets. This paper proposes a maritime target detection and tracking method for the GF-4 satellite image sequence based on the differences in information between frames in the image sequence. First, a preprocessing method is proposed for the unique characteristics of the image. Then, the ViBe (Visual Background Extractor) algorithm is used to extract the targets in the image sequence. After detection, the DCF-CSR (discriminative correlation filters with channel and spatial reliability) is used as a tracker to track and correlate the detected target to complete the task of predicting and monitoring the targets' movements. Finally, through the comparative analysis of experiments with several classic methods, the feasibility and effectiveness of this method are verified.

**Keywords:** geosynchronous orbit satellite; optical remote sensing image; maritime target; target detection and tracking

---

## 1. Introduction

With the advancement of spaceborne remote sensing technology, the data provided by remote sensing satellites have increased dramatically, and the update speed has become faster. The development of automated algorithms that can accurately and punctually extract useful information from massive remote sensing images is a key issue in the application of remote sensing technology. Among them, the detection of targets in satellite remote sensing images has always been a research hotspot in the field of remote sensing image processing.

Remote sensing satellites have become an important means of detecting various ground activities at home and abroad due to their large observation area and periodic repeatable visits. There are many satellites used for remote sensing image observation. For example, there are the "Ziyuan Series Satellites" [1,2] in China. Among them, the resolution of "Ziyuan-I" (ZY-1) reaches 2.36 m, and the "Ziyuan-III" (ZY-3) is a high-resolution stereo mapping satellite that can draw scales. There are also the "Gaofen Series Satellites" [3,4] in China, where the resolution of the full-color image of Gaofen-1 (GF-1) reaches 2 m, and the average revisit period is four days. Foreign remote sensing image satellites include Quickbird [5] in the United States, Worldview-2 [6], and Systeme Probatoire d'Observation de la Terre-5 (SPORT-5) [5] in France. Given the different sensors mounted on the satellite platform, remote sensing images can be divided into infrared remote sensing images [7], visible light remote sensing images [8], multi-spectral remote sensing images [9], and synthetic aperture radar (SAR) images [10]. Although the visible light remote sensing images are greatly affected by factors such as

sunlight and clouds [11], under the conditions of sunny weather and good sea conditions, they can better reflect the shape of the target and are easy for human eyes to recognize. So, target detection using optical remote sensing images has great significance.

In recent years, research on target detection methods based on visible light remote sensing images has attracted much attention. Different target detection methods have been generated according to different types of targets. For example, the authors of [12] compared Faster R-CNN (Faster Region Convolutional Neural Networks), SSD (Single Shot MultiBox Detector), and YOLO-v3 (You only look once v3) based on the latest convolutional neural network (CNN) target detection model evaluation. By using these three methods to process a limited amount of labeled data and automatically detect aircraft in VHR (Very High Resolution) satellite images, the conclusion that Faster R-CNN architecture provides the highest accuracy is obtained. In [13], a detection method for roads is proposed, which uses a new shape descriptor to separate roads from non-road objects with similar spectra and to identify road network intersections. In addition, the authors of [14] proposed a new detection algorithm using motion information, using the correlation of object motion in multiple frames and satellite attitude motion information to detect target objects in space. In addition to these kinds of targets, there are moving targets at sea. In this era of rapid development of maritime traffic and shipping, ships as carriers of sea transportation, there is great practical significance for the detection of their movements. In the study of [15], by comparing and analyzing the reflectance of each channel of the Korean multi-function SATELlite-2 (KOMPSAT-2) image in the area around Gwangyang Bay, a new ship detection index was proposed—spatial reflectivity distribution—to search for missing ships. Bi, F. [16] et al. proposed a novel layering method. First, in the candidate region extraction stage, an omnidirectional intersecting two-dimensional scanning (OITDS) strategy was designed to quickly extract candidates from land–water segmentation image regions. In the candidate area recognition stage, a mixed decision model (DMM) is proposed to differentiate real ships from candidate objects. This method is used for the detection of offshore ships in complex port areas. For ship detection and tracking, Li [17] et al. proposed a method to automatically detect and track mobile ships of different sizes using satellite video. First, motion compensation between two frames is implemented. Then, the saliency maps of the multi-scale difference images are combined to create a dynamic multi-scale saliency map (DMSM), which is more suitable for detecting ships of different sizes. Third, the candidate motion area is segmented from the DMSM, and after eliminating false alarms based on the surrounding contrast, the moving ship can be detected. The detection of these ship targets mentioned above is aimed at high-resolution and low-orbit satellite remote sensing images. These images usually detect a small area, and only focus on ports, coasts, and other places. Although the resolution is higher, the picture frame is limited. Based on this problem, this paper proposes a method of marine target detection and tracking based on geosynchronous satellite visible light remote sensing images.

The remote sensing satellite involved in the method proposed in this paper is the Gaofen-4 (GF-4) geosynchronous orbit satellite. We have proposed a method for the detection and tracking of moving targets in the sea based on its visible light band image sequence. We first preprocess the image sequence to achieve a certain denoising effect, so as to exclude the detection interference of sea surface noise or cloud noise on the target. Then, we use the ViBe (Visual Background Extractor) algorithm [18] to model the image sequence in order to extract the foreground in the sequence, that is, the moving target. Then, through the conversion of geographic location coordinates to achieve the conversion of the target in actual Earth coordinates, the target is completely positioned. Finally, a discriminative correlation filter with channel and spatial reliability (DCF-CSR) [19] is used to correlate and track the target between frames.

The arrangement of the sections in this article is as follows. In Section 2, we mainly introduce the characteristics of the GF-4 satellite and the characteristics of the remote sensing image in the visible light band, which leads to the reason for why we propose such a method. Section 3 mainly introduces the overall process of the proposed method, including image preprocessing, the ViBe algorithm, the actual

geographic target positioning, and the DCF-CSR target tracking algorithm. Section 4 introduces the results and analysis of the target detection experiments. Section 5 is a conclusion of the full paper.

## 2. Characteristics of the GF-4 Geosynchronous Satellite and Its Image Data

Gaofen-4 (GF-4) [4] is an optical remote sensing satellite with high orbit (36,000 km). Equipped with a staring camera with a width of more than 400 km, it has the abilities of visible light, multi-spectral, and infrared imaging. The resolution of visible light and multi-spectral imaging is better than 50 m. Taking the GF-1 low-orbit satellite (orbit height 638 km) as an example, compared to GF-4, the full-color resolution of GF-1 is 2 m, which is much higher than that of GF-4's 50 m resolution, and it has a higher-definition field of view. However, its imaging observation field of view is only 64,000 square kilometers, which is smaller than the 160,000 km$^2$ of GF-4. Moreover, the revisit time for GF-1's fixed area takes four days, which is greater than GF-4's one day, so the time resolution of GF-1 is also lower. Thus, the wide observation range of the GF-4 high-orbit remote sensing satellite (Figure 1a) and short access period can make up for the shortcomings of the low-orbit remote sensing satellite.

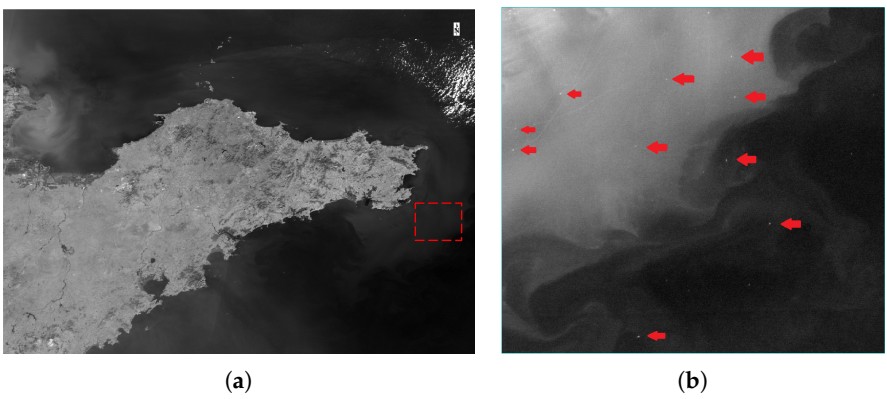

(**a**) (**b**)

**Figure 1.** (**a**) Visible light band image of Gaofen-4 (GF-4); (**b**) enlarged image of the red boxed area in (**a**).

However, due to the low resolution of the GF-4 satellite's remote sensing image, it has brought great challenges to target detection. Because of the imaging on the high orbit, marine targets, such as ships, appear very tiny in images. The targets are in the shape of "small white dots" (shown in Figure 1b, targets are marked with red arrows). Such "white dot-like" targets do not have obvious features, such as shapes and textures, so they cannot be effectively extracted. Moreover, the reflection of sunlight on the waves and clouds [11] on the surface of the sea also appears "dot-like" in the image. There are still static reefs and islands that are also very similar to targets in shape and appearance (Figure 2). Therefore, it is very difficult to directly detect and recognize the target in a single-frame image by using the features of targets. Therefore, it is of great significance to be able to perform target detection in GF-4 images. According to the above characteristics of GF-4 remote sensing images, this paper uses the information differences between GF-4 image sequences to propose a target detection and tracking method that combines image preprocessing, the ViBe foreground extraction algorithm, and the DCF-CSR tracking algorithm to detect and analyze the maritime targets.

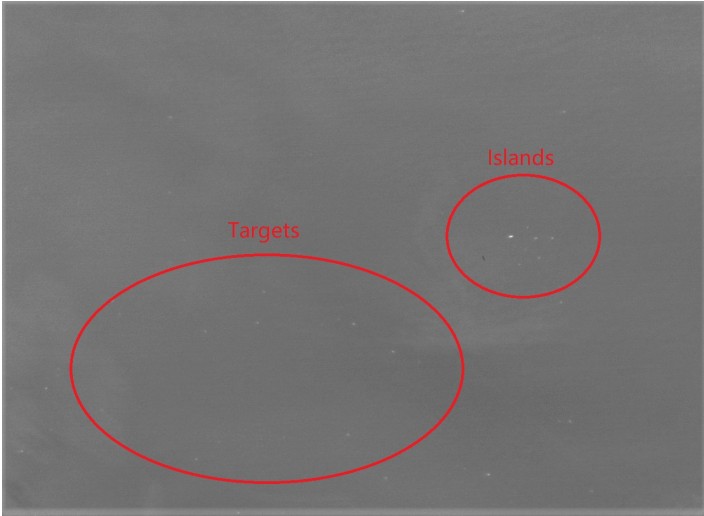

**Figure 2.** Similar islands and targets.

## 3. Detection and Tracking of Maritime Moving Targets

The process of detecting and tracking maritime moving targets in this paper is shown in Figure 3. After correcting the image sequence of the GF-4 satellite through geographic location information, the image is preprocessed to eliminate cloud noise. Through the ViBe background modeling method, we can extract the foreground image, that is, the binary image of the moving targets. Then, we calculate the center of mass of the target to obtain the pixel coordinates of targets. After geographic affine transformation, the real geographic location information of the targets in the world coordinate system can be obtained. Finally, through the DCF-CSR tracking algorithm, the targets in the sequence are tracked one by one and correspondingly associated.

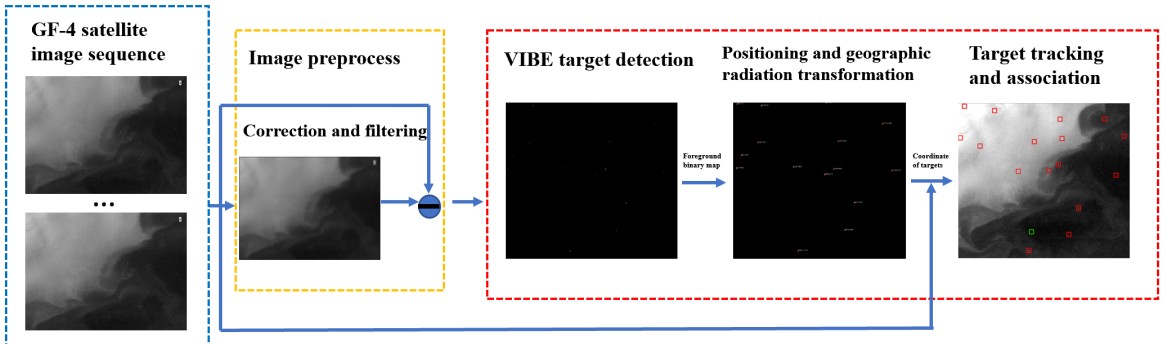

**Figure 3.** The overall process of detection and tracking.

### 3.1. Image Preprocessing

3.1.1. Geolocation Correction of Remote Sensing Images

Due to the fact that the GF-4 satellite is orbiting 36,000 km from the Earth, it is almost a hundred times the height of the low-Earth orbit. If the attitude of the satellite deviates by an angular second, the ground will deviate by several kilometers. Therefore, during the imaging process of the remote sensing satellite, various errors will occur when the sensor is stationary relative to the surface of the Earth. For example, the performance of the sensor itself, the deviation of the technical index from the standard value, etc. are internal errors. External errors are errors caused by unexpected factors of the sensor, such as changes in sensor position and orientation, uneven sensing medium, etc. These errors make it so that the image of the sensor is not on a geographic reference. In order

to prevent these factors from interfering with the target detection of the image sequence, the RPC (Rapid Positioning Coefficient) parameter file is used to geometrically correct the image sequence during the preprocessing stage. The main steps include:

The left side of Figure 4 below is an original image $(abcd)$, defined in the image coordinate system $a - xy$; the right side, $O - XY$, is the map coordinate system, $(a'b'c'd')$ is the corrected image, and $(ABCD)$ indicates the range of the corrected image and the corresponding ground position in the computer.

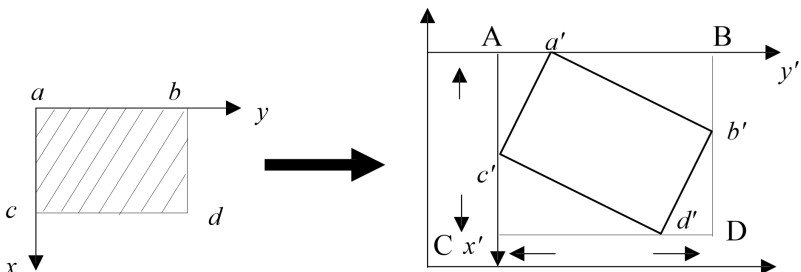

**Figure 4.** The overall process of detection and tracking.

The four corner points $a, b, c, d$ of the original image are transformed into the map coordinate system according to the transformation model, and four coordinate values are obtained: $(X_{a'}, Y_{a'})$, $(X_{b'}, Y_{b'})$, $(X_{c'}, Y_{c'})$, and $(X_{d'}, Y_{d'})$. Then, the minimum values $(X_1, Y_1)$ and maximum values $(X_2, Y_2)$ of these 4 coordinate values are found using $X$ and $Y$ for the two coordinate groups:

$$X_1 = \min(X_{a'}, X_{b'}, X_{c'}, X_{d'}) \tag{1}$$

$$X_2 = \max(X_{a'}, X_{b'}, X_{c'}, X_{d'}) \tag{2}$$

$$Y_1 = \min(Y_{a'}, Y_{b'}, Y_{c'}, Y_{d'}) \tag{3}$$

$$Y_2 = \max(Y_{a'}, Y_{b'}, Y_{c'}, Y_{d'}). \tag{4}$$

Among them, $X_1$, $Y_1$, $X_2$, and $Y_2$ are the map coordinate values of the four boundaries of the corrected image range. The total number of rows and columns of the corrected image can be calculated according to the ground size $\Delta X$ and $\Delta Y$ of the output pixel. Thus, the image range and the image width and height can be obtained.

The geodetic latitude and longitude coordinates $P_0(lon, lat)$ of the image points are calculated according to the image point coordinates $p(r, c)$ of the image to be corrected and the RPC parameters. Given the elevation surface $h$, the image point coordinates $p_0(r, c)$ of the image point on the original image are calculated from the geodetic coordinates $P_0(lon, lat)$ of the object point elevation $h$. According to the bilinear interpolation method, the pixel value of point $p(r, c)$ is interpolated. The pixel values of all the image points to be corrected in the sub-process are calculated in sequence, that is, the image is re-sampled.

### 3.1.2. Mass Speckle Removal

It can be seen from the characteristics of the image of the GF-4 satellite in Section 2 that the cloud noise on the sea surface makes the original tiny targets more close to the background, which makes the detection method using the target features very difficult when using targets based on video sequences. Detection is also prone to a large number of false detections. Furthermore, from the entire sequence, the noise formed by the reflection of these sea waves and the reflection of clouds and fog also shows the phenomena of movements and flickers in the sequence, which are also reasons for the misdetection caused by the method of information difference between frames. Therefore, before the image sequence is detected, image preprocessing needs to be performed to avoid the above situation.

　　　Through the analysis of the grayscale distribution of the Figure 5b (as shown below), the red circle marks the grayscale of the target, showing a peak state. The cloud noise is smoothly distributed in the background. Inspired by the smoothing effect of the mean filter on the image, we pass the image through the mean filter to eliminate the peaks in the image to obtain a smooth image that is a background image similar to noise, as shown in Figure 4. Finally, the background is subtracted from the original image, so as to achieve the effect of eliminating cloud noise, and thus completing the preprocessing of the GF-4 image. The preprocessing method can only filter out the massive cloud speckles, and the distinction between the remaining static islands and the target needs to be completed by the background modeling method.

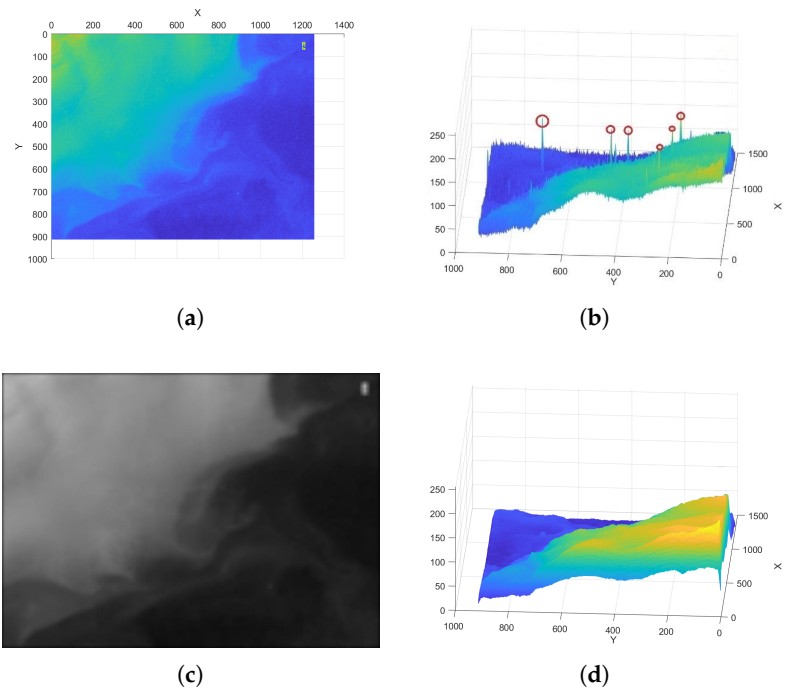

(a)　　　　　　　　　　　　　　　　　　(b)

(c)　　　　　　　　　　　　　　　　　　(d)

**Figure 5.** (**a**) is the grayscale distribution of Figure 1b, (**b**) is the grayscale three-dimensional distribution of Figure 1b, (**c**) is the image after mean filtering, and (**d**) is the grayscale three-dimensional distribution of (**c**). The $x$ and $y$ axes represent the pixel coordinates of the image, and the $z$ axis represents the gray value of the image.

### 3.2. Moving Target Detection

　　　In 2011, Olivier Barnich and Marc Van Droogenbroeck et al. [18] proposed a new background modeling method: The ViBe (Visual Background Extractor) algorithm. It is a general-purpose target detection algorithm, and has no specific requirements for video stream type, color space, and scene content. This algorithm introduces the random selection mechanism into the background modeling for the first time, and describes the random variability of the actual scene by randomly selecting samples to estimate the background model. By adjusting the time sub-sampling factor, very few sample values can cover all background samples, taking into account both accuracy and calculation load. Aiming at the problem that the GF-4 image sequence cannot use the feature extraction method to detect the target, the ViBe algorithm, which is good at detecting moving targets from the image through background modeling, can have a very good effect. The algorithm flow of the Vibe method is divided into three steps, similarly to the general background modeling method: Background modeling initialization, foreground extraction, and background model update. Due to the characteristics of very small targets in GF-4 satellite images, this paper also made special adjustments to the parameters in the ViBe algorithm.

The background model of the ViBe algorithm describes the background model by defining a sample set of each pixel in the image. The *K* samples are expressed as:

$$S^k, k \in [0, k).\qquad(5)$$

Some popular methods, such as [20,21], require a sequence of tens of frames to complete the initialization of the model. ViBe initializes the background model through a single-frame image. The background model is obtained by randomly sampling the pixels around the corresponding pixel using the uniform law, so it is also called the sampling background model, which is a significant advantage of the ViBe algorithm. For the case where the number of GF-4 satellite image sequence frames is small, ViBe can adapt it well. Therefore, the model initialization is described as:

$$S^k_{i,j} = P_{i,j}, k \in [o, K).\qquad(6)$$
$$r = i + rand[-1, 1]\qquad(7)$$
$$c = j + rand[-1, 1],\qquad(8)$$

where *r* and *c* are the corresponding eight neighborhood points, and *i* and *j* are the coordinates of the specified pixel points in the image. The advantage of this method is that it can greatly shorten the background establishment time, and can also learn quickly when the background changes greatly.

The foreground extraction of the ViBe algorithm is performed by calculating the distance $Dist_{i,j}$ (pixel value difference) between the new pixel point $P_{r,c}$ and each sample value $S^k_{i,j}$ in the sample set $S_{i,j}$. When the calculation result is less than the given threshold d, it is considered to be similar to the specified sample. When the number of samples $N$ is greater than $\#_{min}$ (the value range is $[2, K/2]$, generally set to 2), the pixel is considered to belong to the background; otherwise, it is judged to be the foreground. The schematic diagram is shown in Figure 6.

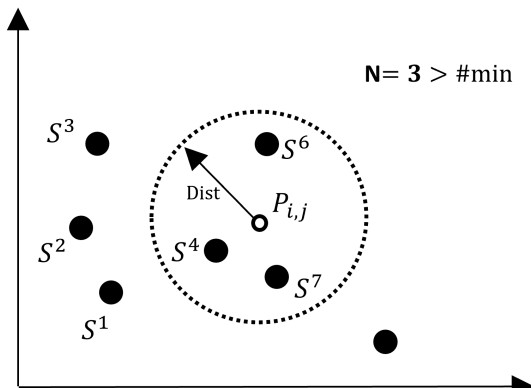

**Figure 6.** Comparison of a pixel value with a set of samples in a two-dimensional Euclidean color space. To classify $P_{i,j}$, we count the number of samples of $S_{i,j}$ intersecting the sphere of radius *Dist* centered on $P_{i,j}$.

Unlike traditional background modeling methods, the ViBe algorithm innovatively proposes a strategy of random selection. Pixels that conform to the background model will participate in the model update strategy. The update strategy is shown in Figure 7. One sample in the corresponding sample set $S_{i,j}$ is randomly selected and replaced with the current value of pixel $P_{i,j}$. Assuming that the learning rate is *LR* (the reciprocal of the update probability; the general value is 2–64; the smaller the value, the faster the update), then the update strategy can be expressed (the flowchart is as follows): (a) When a pixel $P_{i,j}$ is determined to be in the background, it has a probability of $\frac{1}{LR}$ to update its corresponding sample set $S_{i,j}$. When the condition is met, one of the sample values is randomly selected for replacement. (b) At the same time, there is a probability of 1/LR to update the value of its

neighbors. When the probability condition is met, one of the 8 neighbor points $P_{(r,c)}$ ($r$, $c$ is the row and column index) is randomly selected. Then, a sample k in the sample set $S_{i,j}$ corresponding to the neighbor point is randomly selected and replaced with the value of pixel $P_{i,j}$.

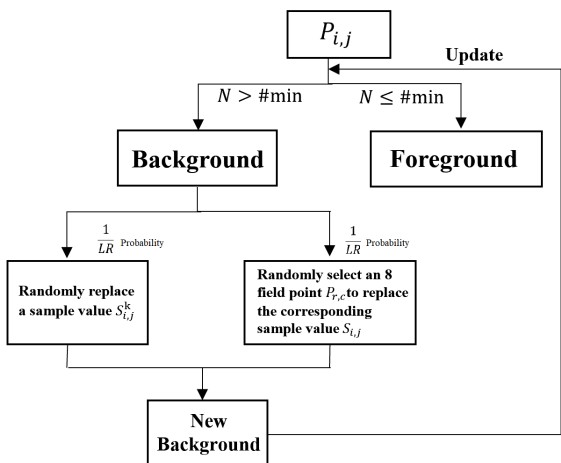

**Figure 7.** Update strategy between every two frames

It is easy to see that in the ViBe model, the model is defined by only two parameters; one is the sample radius R, and the other is the minimum cardinality $\#_{min}$. In [18], the author mentions that ViBe is a non-parametric model, and there is no need to adjust these parameters during background extraction. however, in [22], for more videos of different types of targets, adjusting the parameters is more suitable for the ViBe algorithm. Among these parameters, the background factor is used to update the probability of updating its model. In [18], the parameters of the earliest version of the ViBe algorithm are set as follows: The number of samples per pixel is 20, the search radius is 20, the min index is 2, and the subsampling rate is 16. However, in our actual target detection experiment, through a large number of GF-4 image sequences, we found that because the target is very weak, the detected target result points are few. The number of samples and the value of min for each pixel should be appropriately increased, so as to increase the number of detection targets and expand the contour of the target, which is convenient for subsequent target positioning and centroid calculation.

### 3.3. Georadiological Radial Transformation

After the targets in GF-4 satellite image sequence are detected by the ViBe algorithm, the foreground in the image is extracted. The result is a binary image sequence segmented by foreground and background. So, to automatically detect the coordinates and position of the target in the actual geographic location, we need to locate the pixel coordinates of the target in the image, and then get the coordinates of the target in the world coordinate system through geographic affine transformation.

First, the Canny operator is used to detect the edges in the image sequence to obtain the target contours, and then the central moments of these contours are calculated. For an $M \times N$ image, $f(i,j)$ ($f(i,j)$ is the gray value of the image at point $(i,j)$), and its $p+q$-th geometric moment $m_{pq}$ is:

$$m_{pq} = \sum_{i=1}^{M} \sum_{j=1}^{N} i^p j^q f(i,j). \tag{9}$$

Then, we can get the coordinates of the center of gravity of the target $(\bar{i}, \bar{j})$, where $\bar{i} = \frac{m_{10}}{m_{00}}, \bar{j} = \frac{m_{01}}{m_{00}}$.

After obtaining the pixel coordinates of the target, the geographic location of the target is determined by geographic affine transformation. Radiation transformation uses the following formula

to express the relationship between coordinates and geographic coordinates on a GeoTIFF remote sensing image:

$$X_{geo} = GT(0) + X_{pixel} * GT(1) + Y_{line} * GT(2) \tag{10}$$

$$Y_{geo} = GT(3) + X_{pixel} * GT(4) + Y_{line} * GT(5) \tag{11}$$

$(X_{geo}, Y_{geo})$ represents the actual geographic coordinates corresponding to the coordinates $(X_{pixel}, Y_{pixel})$ on the image. For an image that goes north to south, $GT(2)$ and $GT(4)$ are equal to 0, $GT(1)$ is the width of the pixel, and $GT(5)$ is the height of the pixel. The $(GT(0), GT(3))$ coordinate pair represents the upper left corner coordinates of the upper left corner pixel. Through this radial transformation, we can get the geographic coordinates corresponding to all the pixels on the image.

### 3.4. Target Tracking and Association

In the previous section, the detection of moving targets in the image of the GF-4 satellite was introduced. However, this is just to mark the target detection in a single frame of the image sequence. The motion information of a target in the entire image sequence cannot be correlated. Here, we will introduce the CSR-DCF tracker (CSRT) to track and correlate the targets in the detected image sequence.

The MOSSE (Minimum Output Sum of Squared Error filter) algorithm is a target tracking algorithm proposed by Bolme, D.S. et al. [23] in 2010. The algorithm uses the correlation filter (CF) technology for the first time. The basic idea of CF tracking is to design a filter template and use the template to perform correlation operations with the target candidate area. The position of the maximum output response is the target position of the current frame.The formula is as follows:

$$y = x \otimes w, \tag{12}$$

where $y$ represents the response output, $x$ represents the input image, and $w$ represents the filter template. The correlation theorem is used to convert the correlation into a dot product with a smaller amount of calculation:

$$\tilde{y} = \tilde{x} \cdot \tilde{w}. \tag{13}$$

$\hat{y}$, $\hat{x}$, and $\hat{w}$ are the Fourier transforms of $y$, $x$, and $r$, respectively. The task of correlation filtering is to find the optimal filtering template.

To improve the robustness of the filter template, MOSSE uses multiple samples of the target as training samples to generate a better filter. MOSSE takes the minimum squared error as the objective function, and uses m samples to find the least squares solution:

$$min(\sum_{i=1}^{m} (\hat{x}_i \cdot \hat{w}^* - y_i)^2). \tag{14}$$

We get $\hat{w}$ as:

$$\hat{w} = \frac{\sum_i \hat{x}_i \cdot \hat{y}_i^*}{\sum_i \hat{x}_i \cdot \hat{x}_i^*}. \tag{15}$$

The method of obtaining $\hat{x}$ and $\hat{y}$ is to perform random affine transformation on the tracking box (groundtruth) to obtain a series of training samples $x_i$, which is generated by the Gaussian function; its peak position is at the center of $x_i$. The value of the corresponding Gaussian map position after the affine transformation of the original image center is the $y_i$ of this sample.

The discriminative correlation filters (DCF) method is constantly updated and improved by researchers. Lukežič, A. et al. [19] introduced the concept of channel and spatial reliability into DCF tracking in 2017, providing a learning algorithm CSR-DCF (discriminative correlation filters

with channel and spatial reliability) for its efficient and seamless integration into the process of filter updating and tracking. This method has higher accuracy than MOOSE and KCF (Kernelized Correlation Filters) [24]. It has achieved the most advanced results on VOT2016 (Visual Object Tracking Benchmark 2016), VOT2015 (Visual Object Tracking Benchmark 2016), and OTB100 (Object tracking benchmark).

For the small target in the GF-4 satellite image sequence, this paper uses the CSR-DCF method combined with the ViBe algorithm to locate and track the target, and achieved good results. This method proposes two concepts of spatial reliability and channel reliability [19]. Spatial reliability uses the image segmentation method to calculate the binary constraint mask of the spatial domain through the background color histogram probability and the prior center. The binary mask here is similar to the mask matrix $P$ in CFLB (correlation filters with limited boundaries) [25]. CSR-DCF uses the image segmentation method to select the effective tracking target area more accurately. Channel reliability is used to distinguish the weight of each channel during detection, and the weight is determined by channel learning reliability and channel detection reliability.

The single tracking iteration process of this method is divided into two parts: Localization and updating, as follows. The first is the localization step. Features are extracted from the search area centered on the target's location as estimated in the previous time step, and they are correlated with the learned filter $h_{t-1}$. The object is located by summing the correlation responses, which are weighted by the estimated channel reliability score $w_{t-1}$. As described by Danelljan et al. [26], the proportion is estimated by a single proportional spatial correlation filter. The filter responses of each channel are used to calculate the corresponding detection reliability values $\tilde{w}^{(det)} = \left[ \tilde{w}_1^{(det)}, ..., \tilde{w}_{N_c}^{(det)} \right]^T$ according to the formula:

$$\tilde{w}_d^{(det)} = max \left( 1 - \rho_d^{max2}/\rho_d^{max1}, 0.5 \right),\tag{16}$$

where $\tilde{w}_d^{(det)}$ is the detection reliability of the $d$-th channel and $1 - \rho_d^{max2}/\rho_d^{max1}$ is the ratio between the second and first highest non-adjacent peak in the channel response graph.

Then comes the update step. The training area is centered on the target location estimated in the localization step. The foreground and background histogram $\tilde{c}$ is extracted and updated by an exponential moving average with a learning rate $\eta_c$ (Step 2 in the Update step of Algorithm 1). The foreground histogram is extracted by the Epanechnikov kernel within the estimated object-bounding box, while the background is extracted from the neighborhood twice the size of the object. The spatial reliability map $m$ is constructed and the optimal filter $\tilde{h}$ is calculated by optimizing the augmented Lagrangian multiplier [27]. Each channel's learning reliability weights $\tilde{w}^{(lrn)} = \left[ \tilde{w}_1^{(lrn)}, ..., \tilde{w}_{N_c}^{(lrn)} \right]^T$ are estimated from the correlation response:

$$\tilde{w}_d^{(lrn)} = max(f_d * h_d),\tag{17}$$

where a discriminative feature channel $f_d$ produces a filter $h_d$ whose output $f_d * h_d$ nearly exactly fits the ideal response.

Next, the reliability weights $\tilde{w}$ of the current frame are calculated according to the reliability of detection and learning:

$$\tilde{w}_d = \tilde{w}_d^{(det)} \cdot \tilde{w}_d^{(lrn)}.\tag{18}$$

The filter and channel reliability weights are updated by an exponential moving average with the learning rate (current frame and starting from the previous frame) (Steps 7 and 8 in the Update step of Algorithm 1).

---

**Algorithm 1** The CSR-DCF tracking algorithm.

---

**Require:**

Image $I_t$ object position on previous frame $p_{t-1}$, scale $s_{t-1}$, filter $h_{t-1}$, color histograms $c_{t-1}$, channel reliability $w_{t-1}$.

**Ensure:**

Position $p_t$, scale $s_t$, and updated models.

**Localization and estimation:**

　1: New target location $p_t$: Position of the maximum in correlation between $h_{t-1}$ and image patch
　　　features $f$ extracted on position $p_{t-1}$ and weighted by the channel reliability scores $w$.
　2: Using per-channel responses, estimate detection reliability $\tilde{w}_d^{(det)}$.
　3: Using location $p_t$, estimate new scale $s_t$.

**Update:**

　1: Extract foreground and background histograms $\tilde{c}^f$ , $\tilde{c}^b$.
　2: Update foreground and background histograms $c_t^f = (1 - \eta_c)c_{t-1}^f + \eta_c\tilde{c}^f$, $c_t^f = (1 - \eta_c)c_{t-1}^b + \eta_c\tilde{c}^b$.
　3: Estimate reliability map $m$.
　4: Estimate new filter $\tilde{h}$ using $m$.
　5: Estimate learning channel reliability $\tilde{w}^{(lrn)}$ from $h$.
　6: Calculate channel reliability $\tilde{w} = \tilde{w}^{(lrn)} \odot \tilde{w}^{(det)}$.
　7: Update filter $h_t = (1 - \eta)h_{t-1} + \eta\tilde{h}$.
　8: Update channel reliability $w_t(1 - \eta)w_{t-1} + \eta\tilde{w}$.

---

## 4. Experiments and Analysis

### 4.1. Preprocessing

　　　According to the preprocessing method introduced in Section 3.1.1, after geolocation correction of the image sequence, the image sequence is corrected according to the geographic location. Then, we use a 20 × 20 mean filter template to filter each frame $I_0$ in the GF-4 satellite image sequence to obtain $I_{mean}$. Because the mean filter is used, the gray value of the cloud part of $I_{mean}$ will be higher than $I_0$. Therefore, using $I_{mean}$ as a mask, the portion of $I_0$ that is smaller than the grayscale of $I_{mean}$ is masked, that is, the "spot block" noise is removed, and the processed image is obtained.

　　　As can be seen from the following preprocessed images and their three-dimensional distribution of grayscale (Figure 8), the mean filter can filter out the target points that show peak states, and it is easy to remove the sea wave reflection and cloud noise with a relatively smooth background through calculation. A small amount of pixel loss of the target will not adversely affect the subsequent target detection, but excludes the interference of noise.

### 4.2. Detection Results and Comparison

　　　After preprocessing the overall image sequence, the ViBe algorithm is used to extract the foreground of the image sequence. The image sequence used in the following experiments is the 1A-level image of the GF-4 geosynchronous orbit satellite and the grayscale image of the visible light band. The actual sea area is on the east side of the Shandong Peninsula of the Yellow Sea in China, and the target is a maritime sport target. The imaging period is 3 min, and the sequence of the group consists of five frames. In the experiment, the ViBe detection algorithm (without preprocessing) and preprocessing + ViBe (pre-ViBe) algorithm were used to detect the target. In addition, we also introduced three other classic video-sequence-based moving target detection algorithms to participate in the comparison, which are the frame difference (FD) method [28], Gaussian mixture model (GMM) method [29], and optical flow (OF) method [30]. In the moving target detection algorithm based on

the video sequence, the first frame of the image is completely involved in the initialization of the background, not as the result frame, so the experimental result is the last four frames. The parameters of the ViBe algorithm are set as follows: The number of samples per pixel is 40, the search radius is 40, the $\#_{min}$ index is 5, and the sub-sampling rate is 16. Figure 9 shows the experimental results.

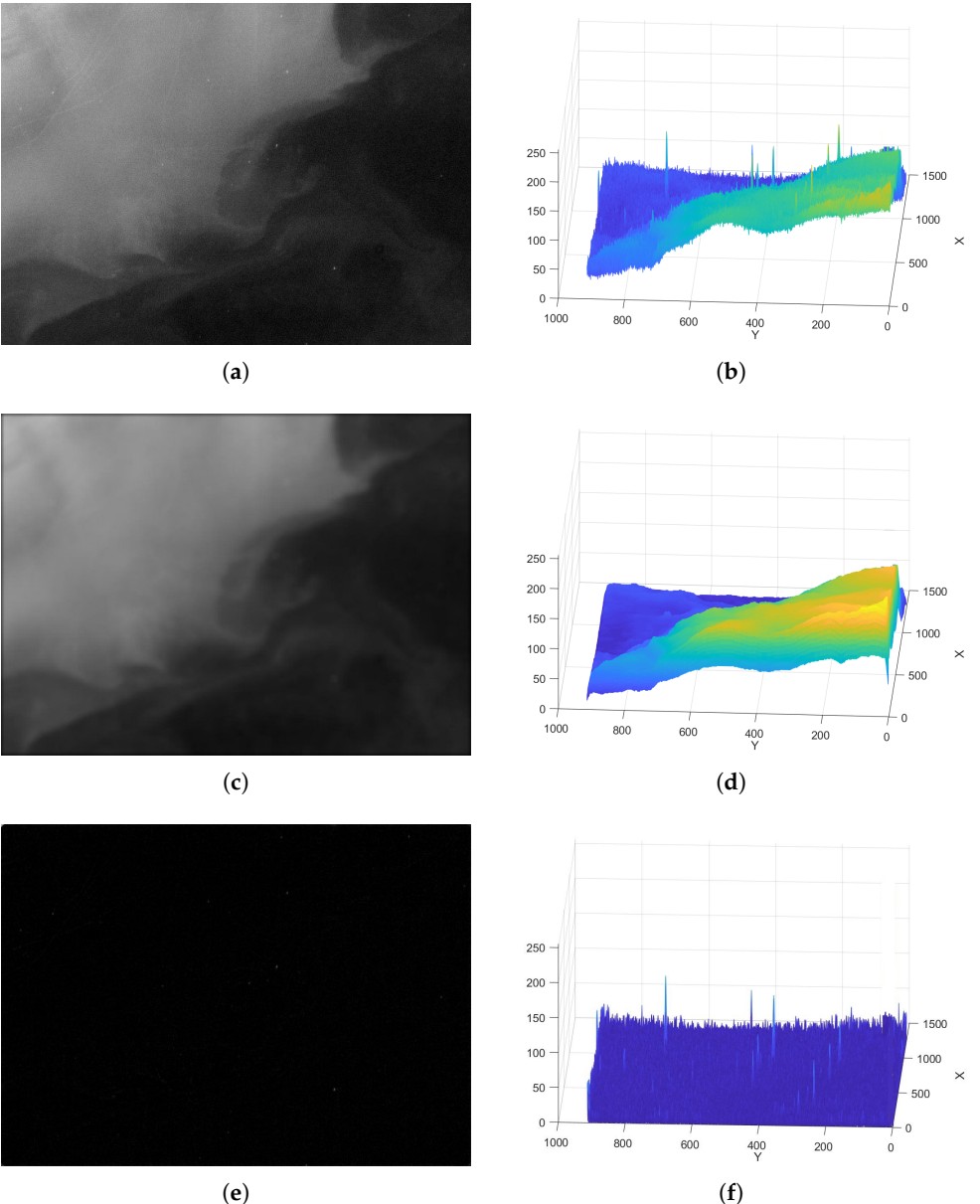

(a)

(b)

(c)

(d)

(e)

(f)

**Figure 8.** (**a**,**b**) are the original image and its 3D gray distribution map; (**c**,**d**) are the filtered image and its 3D gray distribution map; (**e**,**f**) are the preprocessing result map and its 3D gray distribution map.

In order to better observe the detected target, we placed a partially enlarged image of pre-ViBe fourth in Figure 10. It can be seen from the comparison of the experimental result images that even if the pre-ViBe method detects the false alarm caused by the false shadow effect, it can still detect a larger number of positive sample targets than the ViBe method without preprocessing. The false shadow effect is a common problem in detecting moving targets with background difference methods. When modeling the background initialization, the moving targets may be in the background, and they will produce false shadows after moving. Compared with the proposed method, the FD method cannot suppress the noise caused by the reflection of light on the sea surface and clouds, and cannot determine the moving target in it. The GMM, which also belongs to the background modeling method, also cannot

suppress the rapid change of the background in the short sequence until the fifth frame's background extraction is slightly improved, but the target cannot be completely extracted. Although the optical flow method can suppress jitter and noise, due to the discontinuity of the target in the image, the optical flow method easily loses the target and the effect is not good.

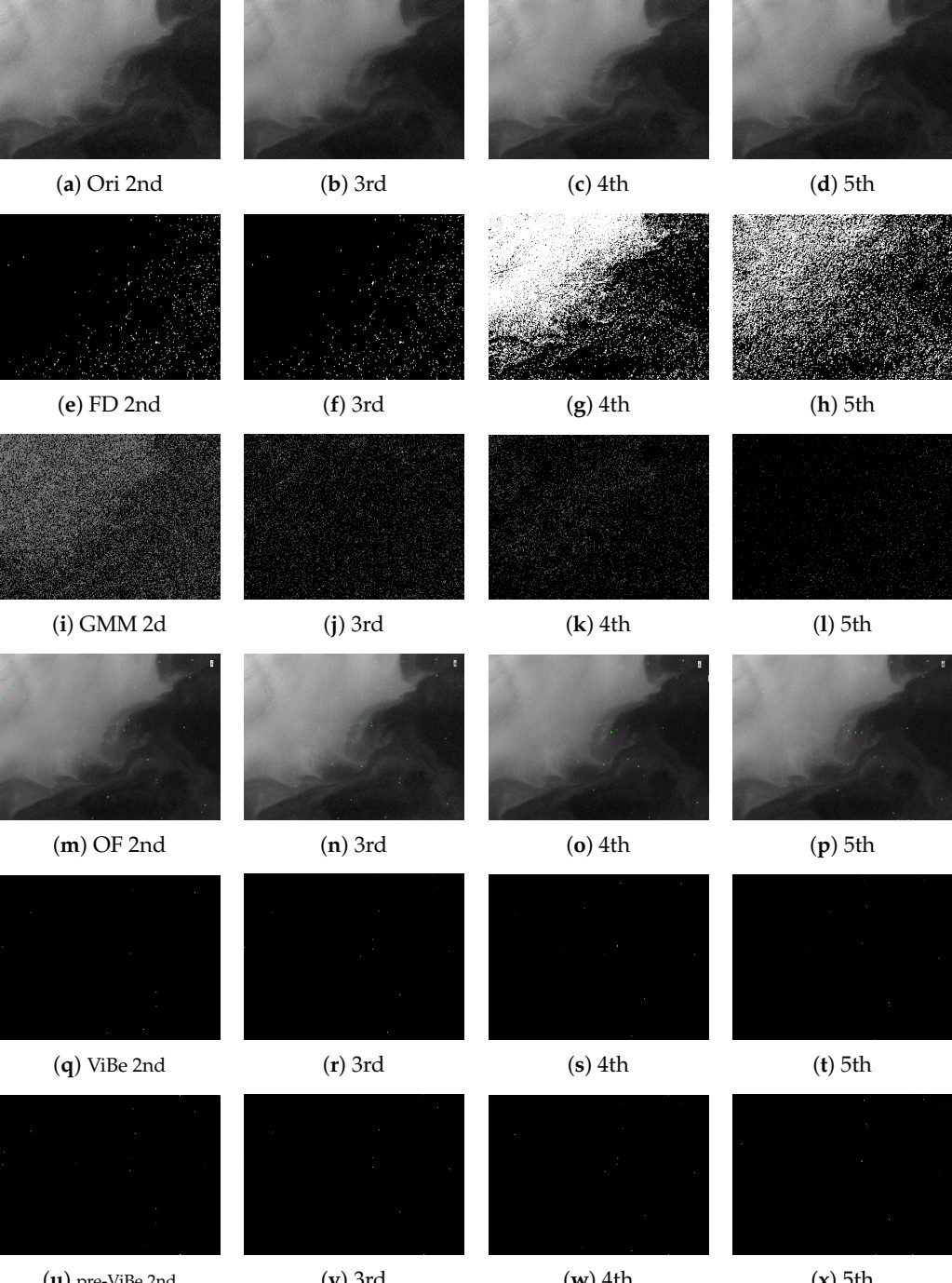

**Figure 9.** From top to bottom are the original image, the results of frame difference (FD), Gaussian mixture model (GMM), optical flow (OF), Visual Background Extractor (ViBe), and preprocessing + ViBe (pre-ViBe); from left to right are the second frame to the fifth frame of the image sequence (one frame per 3 min).

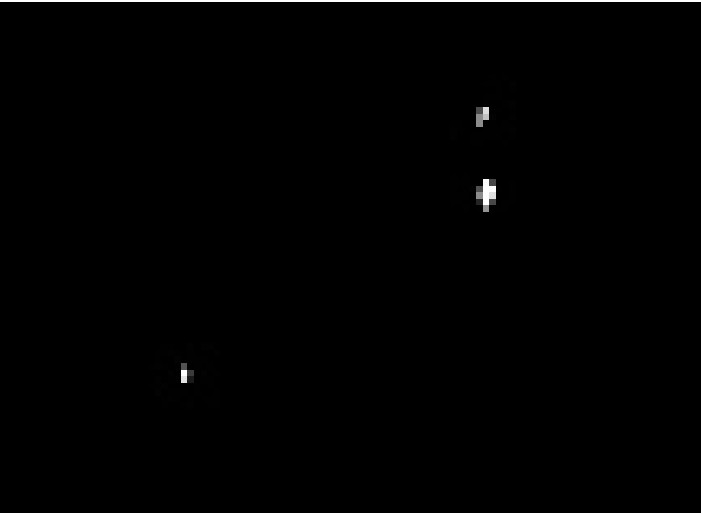

**Figure 10.** Enlarged view of the target the fourth frame of pre-ViBe method (108 × 78).

In order to better reflect the ability of proposed target detection, we use another set of long sequences to conduct experiments and compare with the algorithms mentioned above. Using long sequences can also show the ability of the remaining algorithms, making the comparison more reasonable. This sequence (as shown in the figure below) is a twelve-frame image sequence with an imaging period of 3 min. Among them, the targets marked by the red boxes in the original picture are the manually marked targets (as groundtruth), the green boxes mark the successfully detected ship targets, and the yellow boxes mark the false detection targets. Since the image sequence itself carries the actual geographic coordinates, it is easy to exclude the land part in the upper left corner and only mark the target at sea. Because the experimental result is a dynamic video stream detected in real time, the land part in the upper left corner is relatively still. To reflect the practical applicability of the pre-ViBe method, it is not considered as a disturbing factor for ship target detection. The experimental results are shown in Figure 11.

Then, the result of manually marking the target as groundtruth is used to calculate the corresponding detection accuracy of each algorithm. The detection accuracy of all frames is calculated and averaged to get the final accuracy. Similarly, we can also calculate the probability of detection and false alarm rate. The accuracy, the probability of detection, and the false alarm rate of a single frame are calculated according to the following formula:

$$P = \frac{TP}{TP + FP}, \quad D = \frac{TP}{GT}, \quad F = \frac{FP}{TP + FP}. \tag{19}$$

Among them, $P$ is the accuracy rate, $D$ is the probability of detection, $F$ is the false alarm rate, $GT$ is the number of groundtruths, $TP$ is the number of samples that predict the positive class as the positive class, that is, the number of correct detection targets, and $FP$ is the number of samples that predict the negative class as the positive class, that is, the number of false detection targets. At the same time, we also recorded the calculation time of each algorithm for 12 frames of images. The configuration of the computer we use is i7-8700 cpu and 16 G memory. The results of experiments above are presented in Table 1.

As can be seen from Figure 11 and Table 1, in the target detection of the twelve-frame sequence, pre-ViBe is superior to other algorithms in accuracy, probability of detection, and false detection rate. In terms of calculation time, since pre-ViBe adds a preprocessing part, the calculation time is slightly longer than ViBe, but it is better than other algorithms. Due to the complex situation of GF-4 satellite images, the GMM method cannot suppress the rapid change of the background, cannot accurately extract the moving targets, and its accuracy cannot be calculated. Based on the above two experiments, the proposed method has a good performance on target detection in such a short sequence, and can

well meet the task requirements of detecting moving targets at sea in a short time. The performance is stable in long sequences, and the accuracy is higher than in other methods based on video sequences.

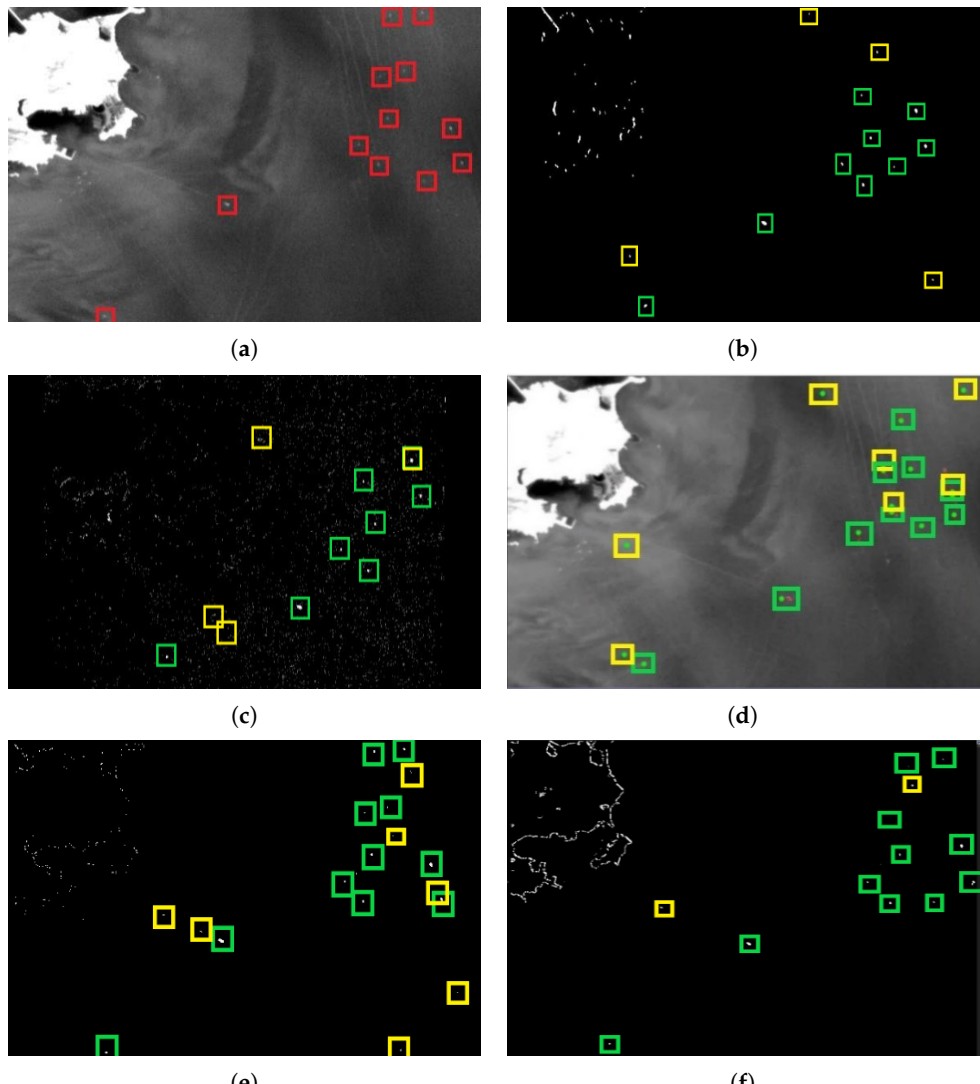

**Figure 11.** (**a**–**f**) are the original image, FD, GMM, OF, ViBe, pre-ViBe results. Take one of the best frames for comparison. In the figure, the red boxes mark the manually marked targets (groundtruth), the green boxes mark the successfully detected targets, and the yellow boxes mark the false detection targets.

**Table 1.** Performance of each algorithm.

| Methods | FD | GMM | OF | ViBe | pre-ViBe |
|---|---|---|---|---|---|
| Accuracy (%) | 69.2 | — | 58.8 | 61.1 | 84.6 |
| Probability of detection (%) | 75 | — | 83.3 | 91.6 | 91.6 |
| False alarm rate (%) | 31 | — | 41 | 38.9 | 15.4 |
| Calculation time of 12 frames (s) | 24.32 | 99.01 | 12.51 | 8.18 | 15.28 |

### 4.3. Positioning

After the foreground image of the image sequence is obtained, according to the centroid detection algorithm mentioned in Section 3.3, the centroid coordinates of the binary connection area (that is, the target) in the foreground are calculated. Through geographic affine transformation, the pixel coordinates of the target are mapped to the geographic location coordinates to obtain the latitude and longitude information of the target. The result of the centroid calculation is shown in the Figure 12 below. This is beneficial for quickly locating moving targets at sea and predicting the movement of the targets, as well as tracking and monitoring.

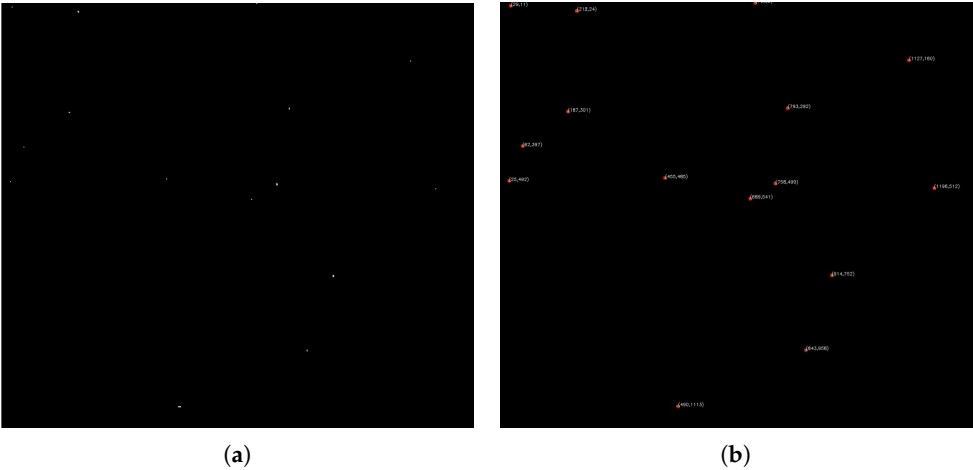

(**a**)          (**b**)

**Figure 12.** (**a**) is foreground map obtained from pre-ViBe; (**b**) is the result of centroid detection. It can be seen from the figure that we have located the coordinates of all the targets in the image.

### 4.4. Tracking

In order to better predict the moving direction of the target and to complete the tracking and correlation of the targets between the frames, the CSRT is used in the tracking process. We use the pixel coordinates of the target after positioning (Section 4.3) as the input of the CSRT, and instruct the tracker to box mark the target groundtruth as the initial template. The space-based binary constraint mask is calculated by the front background color histogram probability and the center prior, and then the channel reliability mentioned in Section 3.4 is used to distinguish the weight of each channel during detection. Finally, the position of the target in the next frame image is calculated, and iterative update is entered. Prior to this, we have obtained the coordinates of all the targets in each frame of the image sequence through ViBe and positioning. By comparing the tracking result of a certain target of interest with the target coordinates in each frame after the current frame, the targets in the sequence can be related one by one.

The image used in the experiment is a twelve-frame-long sequence. We take the tracking results of the second frame, the fifth frame, and the 11th frame, as shown in Figure 13 below. In this long sequence, the CSRT performs very well and does not lose the target. After tracking the CSRT of the sequence, we can obtain the position information of each interest target point in a sequence, which can well predict the moving direction of a moving target in the sea, and can also calculate the average speed of the target and other information. It is of great help for the tracking and surveillance of moving targets at sea.

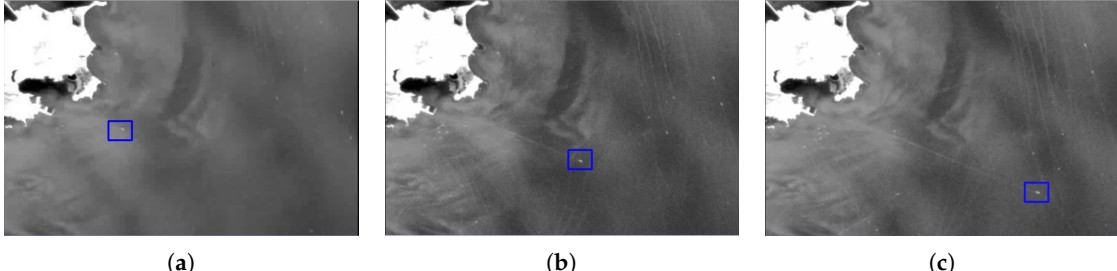

|              |              |              |
|:------------:|:------------:|:------------:|
| (**a**)      | (**b**)      | (**c**)      |

**Figure 13.** From left to right are the (**a**) second frame, (**b**) fifth frame, and (**c**) 11th frame of the image sequence, respectively.

## 5. Conclusions

Because the GF-4 satellite can keep synchronous operation with the Earth, it can observe a certain area for a long time; the access period is short and the frame is large. Using it to observe the Earth's surface has obvious advantages over other near satellites. However, the actual spatial resolution and the unique characteristics of the images have brought great difficulties for target detection using its images. Based on this background, this paper proposes a target detection and tracking method based on the difference of information between frames in the image sequence, which contributes to the detection of moving targets on the sea based on the GF-4 satellite's remote sensing images. The proposed method consists of preprocessing, foreground extraction based on ViBe, a centroid detection and positioning method, and a target tracking algorithm based on CSRT. Through the experimental analysis in Section 4, we find that our preprocessing can effectively remove blocky spots and reduce their interference in target detection. Compared with the detection results of several classic video-sequence-based algorithms, the ViBe algorithm is more suitable for the GF-4 remote sensing satellite images, and its accuracy and stability are better than those of other algorithms. In the final target tracking experiment, we use CSRT to track the detected target. The tracking effect of this method is very stable when the sequence is long and the target is discontinuous. The above experimental analysis also proves the feasibility and effectiveness of the proposed method.

In future work, we will continue to explore the characteristics of GF-4 remote sensing images. Due to the interference of clouds and reflections on the sea surface, target detection is very difficult. We will continue to explore the work of cloud removal and denoising of such images. At the same time, we will optimize the target detection algorithm and improve the task of predicting the target movement during target monitoring and tracking; for example, by measuring the target speed and moving direction. Finally, we hope to combine the proposed algorithm with hardware to form a complete automatic identification system that has greater significance for target detection in GF-4 satellite images.

**Author Contributions:** The authors of this article have made important contributions to the research ideas or design. And the authors obtain, analyze, and interpret data for research. All authors have read and agreed to the published version of the manuscript.

**Funding:** The National Natural Science Foundation of China (61571377, 61771412, 61871336) and the Foundamental Research Funds for the Central Universities (20720180068).

**Conflicts of Interest:** The authors declare no conflict of interest.

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
