# Peer review of "Detection and Tracking Method of Maritime Moving Targets Based on Geosynchronous Orbit Satellite Optical Images"

_electronics, doi:10.3390/electronics9071092_

Round 1

Reviewer 1 Report

Thanks for this article about maritime target detection using time series of GF-4 remote sensing images.

Summary

The article starts by explain the needs of detection using light bands. Then it describes the characteristics of GF-4. Due to the low resolution, ships and islands look similar. The detection is the result of set images comparison to determine the background. The article details coordinates issues. Then it explains the visual background extractor method. The article gives some visual results and a table of accuracy to compare with some other methods.

28 references (25% since 2018, 25% before 2010)

Strengths

The detection is based on the relative homogeneity of the sea surface. The method seems to fit with the GF-4 sensor resolution.

Weakness

Detection adresses only targets with same range size (not too low, not too high).

ViBE implementation and coordinate conversion not sound as novel methods in the article.

No future works are descripe.

Comments

Equations (7) and (8) suggest the extraction depends of the resolution. What about high resolution? In that case, a ship can be considered as background itself.

The article is not clear about efficiency.

How many loops in figure7? What is the algorithm complexity? What the computation time with which hardware configuration?

Why don't enforce the matching with other informations (as Automatic Identification System)?

What about optimal transport to associate changing values (targets) between frames?

L25: Why mention ration 1:50000? Ratio makes sense only with physical support and the article is a methodological (virtual) approach.

L28: “SPORT-5”. Do you mean “SPOT-5” (SPOT stand for Satellite Pour l’Observation de la Terre)?

L39: Please indicate “Very Hight Resolution” before using VHR.

L45: what “era” means? (area).

L83: The terms “in general”, “is smaller”, “is lower” not sounds rigorous. With what is it compared? Please give values and facts.

L88: Figure 1 (b). Please explain the red arrow in the caption.

L88: “No. 4” is not clear in the text. Does it a reference to GF-4?

L204: Figure7, under right bottom corner of “background” box, the text is not readable in latin.

L222: “target contours”. In the article (and all figures) the contours seems to be limited as only one pixel. Could you use “contours” in that case?

L310: Figure8(c). It is curious that the North direction in the upper right corner is treated is real pixel and filtered. Why don't use original image? It appears again in Figure8(e). But it must be remove by invariant property during the filtering.

L325: Figure9. The caption indicates “the 2nd to the 5th frame”. Please repeat 3 minutes interval here.

L325: Figure 9. We can't see details. Please give apart an enlarged region to highlight the target contours of one.

L384: The conclusion gives no perspective or future works.

Author Response

Response to the Reviewer 1 Comments:

General comments:

Comments and Suggestions for Authors

Thanks for this article about maritime target detection using time series of GF-4 remote sensing images.

Summary

The article starts by explain the needs of detection using light bands. Then it describes the characteristics of GF-4. Due to the low resolution, ships and islands look similar. The detection is the result of set images comparison to determine the background. The article details coordinates issues. Then it explains the visual background extractor method. The article gives some visual results and a table of accuracy to compare with some other methods.

28 references (25% since 2018, 25% before 2010)

Strengths

The detection is based on the relative homogeneity of the sea surface. The method seems to fit with the GF-4 sensor resolution.

Weakness

Detection adresses only targets with same range size (not too low, not too high).

ViBE implementation and coordinate conversion not sound as novel methods in the article.

No future works are descripe.

Response: First of all, thank you very much for your comments on this paper. We have carefully read your comments and think that your comments are very helpful for us to modify and improve this article. Based on your suggestions, we have made point-to-point modifications in the article. And the modified parts of the manuscript are marked with colors. We hope that our reply and revision will meet your requirements. We believe that we have unquestionably improved the quality of this paper with the amendments. Sincere thanks again!

Point 1:Equations (7) and (8) suggest the extraction depends of the resolution. What about high resolution? In that case, a ship can be considered as background itself.

Response 1: First of all, thank you very much for your comments. Next let me explain your doubts. Due to the low resolution of the GF-4 satellite optical image, the target is tiny and has no features such as shape and texture. Therefore, it is very difficult to detect the target in this case. But in the image sequence, the human eye can find the moving target. According to such image characteristics, we have proposed the idea of using image sequence background extraction to process GF-4 images. So if it is a high-resolution image, it is not suitable for this kind of method. High-resolution images contain many feature information of the target, and may use deep learning-based methods, such as YOLO, RCNN, etc.

Point 2:The article is not clear about efficiency.

Response 2: Thank you for your suggestion. Regarding efficiency, I will reply to your comments in several ways. The first is the shooting efficiency of GF-4 single image. It takes 3-12 minutes to image an image covering 160000 square kilometers. According to the transmission rate of 300 megabits per second of the GF-4 satellite, a photo can be taken in 3-4 seconds. Then comes the proposed algorithm efficiency. The detection time of the pre-ViBe algorithm for the 12-frame image sequence is 15.28 seconds, and the average detection time of one frame is about 1.27 seconds (this part we added in Table1.)

Point 3:How many loops in figure7? What is the algorithm complexity? What the computation time with which hardware configuration?

Response 3: Thanks for your question. In Figure.7, the background update process of a single-frame image is shown. It is one loop between two frames. The detection time of the pre-ViBe algorithm for the influence sequence of 12 frames is 15.28 seconds, and the average detection time of one frame is about 1.27 seconds. The computer used is configured with i7-8700 cpu and 16 g memory. These contents have been supplemented in the text (Figure.7, L364 and Table.1).

Point 4:Why don't enforce the matching with other informations (as Automatic Identification System)?

Response 4: Thank you for your suggestion. We very much agree with you. It is a very good idea to implement our proposed target detection and tracking method into an effective automatic identification system. At present, although our method has had a certain effect, but only at the algorithm level, it has not been combined with hardware to build a complete engineering system. These are the key points of our future work.

Point 5:What about optimal transport to associate changing values (targets) between frames?

Response 5: Thank you very much for your question. For optimal transmission between frames, the ViBe algorithm has the following characteristics in the background model update strategy.

1). Memoryless update strategy: every time it is determined that the background model of a pixel needs to be updated, a sample value of the pixel sample set is randomly replaced with a new pixel value.

2). Time sampling update strategy: Not every frame of data needs to be updated, but the background model is updated at a certain update rate. When a pixel is determined as the background, it has a probability of 1/LR to update the background model. LR is the time sampling factor, which is generally 16.

3). Spatial neighborhood update strategy: For pixels that need to be updated, randomly select a background model for the neighborhood of the pixel, and update the selected background model with new pixels.

The steps of the specific update method: each background point has a probability of 1/LR to update its own model sample value, and also has a probability of 1/LR to update its neighbor's model sample value. Updating the sample values ​​of neighbors takes advantage of the spatial propagation characteristics of pixel values, and the background model gradually diffuses outward, which is also conducive to the faster identification of false shadow areas. At the same time, when the current scenic spot count reaches a critical value, it becomes the background, and there is a 1/LR probability to update its own model sample value.

Point 6:L25: Why mention ration 1:50000? Ratio makes sense only with physical support and the article is a methodological (virtual) approach.

Response 6: Thank you for this comment. The mention of "ration" does deviate from the meaning of the article. We have deleted this sentence in L24.

Point 7:L28: “SPORT-5”. Do you mean “SPOT-5” (SPOT stand for Satellite Pour l’Observation de la Terre)?

Response 7: "SPORT-5" here refers to one of the SPOT series satellites developed by the French Space Research Center (CNES). So far, SPOT satellites 1-7 have been launched. "SPOT" is the abbreviation of "Systeme Probatoire d’Observation de la Terre" in French, meaning Earth Observation System. We have added the full name of SPOT-5 to L27.

Point 8:L39: Please indicate “Very Hight Resolution” before using VHR.

Response 8: Thank you very much for your correction. Here is our mistake, we have added in the article L40.

Point 9:L45: what “era” means? (area).

Response 9: Thank you very much for your comments. Here "era" means epoch, period.

Point 10:L83: The terms “in general”, “is smaller”, “is lower” not sounds rigorous. With what is it compared? Please give values and facts.

Response 10: Thank you very much for your comments. Here taking GF-1 low-orbit satellite (orbit height 638 km) as an example, compared to GF-4, the full-color resolution of GF-1 2 m is much higher than that of GF-4’s 50 m resolution, and it has a higher-definition field of view. However, its imaging observation field of view is only 64000 square kilometers, which is smaller than the 160,000 square kilometers of GF-4. Moreover, the revisit time for GF-1's fixed area takes 4 days, which is greater than GF-4's 1 day, so the time resolution of GF-1 is also lower. We have made changes in article L86-91 based on your suggestions.

Point 11:L88: Figure 1 (b). Please explain the red arrow in the caption.

Response 11: Because the target is small, the red arrow is used here to mark the location of the target in the figure (if you zoom in the image, you can see the tiny target). This makes it easier for readers to see the target. We explained it in L95.

Point 12:L88: “No. 4” is not clear in the text. Does it a reference to GF-4?

Response 12: Here is our writer error. We have made changes to L93 in the text. Change the original sentence to: "the GF-4 satellite remote sensing image".

Point 13:L204: Figure7, under right bottom corner of “background” box, the text is not readable in latin.

Response 13: Thanks for your comment. Here is our editorial error. We have modified Figure.7 in the text.

Point 14:L222: “target contours”. In the article (and all figures) the contours seems to be limited as only one pixel. Could you use “contours” in that case?

Response 14: Thank you very much for your comments. In the original image, the target may be only a few pixels. But after ViBe algorithm and edge detection, before contour detection, the pixels in the target area can already reach a certain number. This is because ViBe's foreground extraction mechanism causes the outline of the target to become larger. So the word "contour" is used here. The details of the target outline can be seen in the response of point 17.

Point 15:L310: Figure8(c). It is curious that the North direction in the upper right corner is treated is real pixel and filtered. Why don't use original image? It appears again in Figure8(e). But it must be remove by invariant property during the filtering.

Response 15: Thank you very much for your correction. Here is the wrong image we placed. Now we have replaced Figure.8 with a new image according to your suggestion.

Point 16:L325: Figure9. The caption indicates “the 2nd to the 5th frame”. Please repeat 3 minutes interval here.

Response 16: Yes, it is imaged every 3 minutes. We have made supplementary explanations in the legend of Figure.9 in the text.

Point 17:L325: Figure 9. We can't see details. Please give apart an enlarged region to highlight the target contours of one.

Response 17: Thanks for your comment. Based on your suggestions, we have added Figure10, in the text. It is an enlarged view of pre-ViBe 4th. There is some loss of images due to typesetting. The enlarged view is as follows:

The enlarged image of pre-ViBe 4th in Figure.9

Point 18:L384: The conclusion gives no perspective or future works.

Response 18: Thank you very much for your suggestions. We added the content of future work in the conclusion to make the conclusion more complete.

Reviewer 2 Report

This paper is concerned with the problem of detecting and tracking ships in GF-4 imagery. The usage of English is stilted, and the paper should be reviewed by someone who is comfortable with English. In my review, I point out a few of the more awkward constructions and misused words. Despite those issues, the presentation of the research is fairly clear, and the paper would be suitable for publication with minor revisions. However, I would like the authors to clarify exactly what is new in this paper. It sounds like most of the methodology was already developed – is any of it new to this paper? I assume this is the first time the authors applied these methods to GF-4 data? Has similar work been published using GF-4 data?

“Particularity” (Line 2, 5, 88, 388) This sounds strange to a native English speaker. I recommend changing it to something like “unique characteristics”. But there is another issue as well. The paper makes it sound like there is something wrong with the camera on GF-4. It sounds like the “particularity” has to do with the imagery. However, I think what the authors intend to highlight is that detecting ships with 50 m resolution imagery is a difficult task – so it is the use of this imagery for this application that gives this problem its unique characteristics (more than the imagery itself).

Line 14, “aerospace” I recommend you either delete this word or replace it with “spaceborne”.

Line 15-16, “How to make the computer” I get what you are trying to say, but that is awkward. Recommend changing it to something like “Development of automated algorithms to” or “Automating the process of”.

Line 28, “According to the different sensors” This is the wrong usage of “according to”, which you could replace with “Given”.

Line 38, Shouldn’t “Evaluation” be “evaluation”?

Line 50, “proposed Spatial reflectivity distribution to search” was confusing, recommend: “proposed, Spatial reflectivity distribution, to search”

Line 75, results are Section 3, not 4

Line 88, “No. 4” What does that mean? Is it the fourth image channel?

Line 99, “takes about 60” What are the units?

“geographic radiation transformation” (Line 112, 221, 227, 362) I really don’t know what you mean by this.

Line 116, “geometric correction” If this is the language you prefer to use, that is fine, but I don’t think “correction” is really the right idea. There is nothing wrong with the images that needs to be corrected, they simply need to be geolocated. Factors such as sensor pointing aren’t really “errors”, but are parameters used to geolocate pixels in an image.

Line 117 “3600M” Don’t you mean “36000 km”?

Line 123 “sensor position and posture”. “Posture” is an odd word to use – I think you mean “pointing” or “orientation”.

Lines 128-130: something is screwed up with line numbering here, but in the line after 128, you have a “The” with no period before it.

Line 164 “volatility” – I get the idea, but you really mean “variability”.

Figure 7 – Do you intend to have those Chinese characters in the figure?

Line 206-207, In these two sentences, it sounds like you are arguing both that parameter adjustment isn’t needed, and that it is. What exactly do you mean by “for more video sequences”? Do you mean that for this particular problem, the default values aren’t optimal? Or do you mean that different sequences need different parameters? Please clarify exactly what you are getting at.

Line 214, It would be nice to either give the values that you are using here, or reference where in the paper those values are.

Figures 9 and 10: It would be really helpful if the panels had labels saying what they represent, e.g., “(a) Original”, … At least do this for the leftmost column.

Line 326: What is the “ghost effect”? This is the first time it is mentioned.

Table 1. Accuracy is fine, but it combines the effects of true detections and false detections. Could you please also give the probability of detection and false alarm rate?

Figure 10: Please say what the colors represent in the figure caption.

Line 367: Change spelling from “Traking” to “Tracking”

Line 386: “access period is short” What do you mean by “access period”? Is that the latency in how long it takes between when the image is taken and when it is available for processing? Is it the length of time that the camera captures the scene?

Author Response

Response to the Reviewer 2 Comments:

General comments:

This paper is concerned with the problem of detecting and tracking ships in GF-4 imagery. The usage of English is stilted, and the paper should be reviewed by someone who is comfortable with English. In my review, I point out a few of the more awkward constructions and misused words. Despite those issues, the presentation of the research is fairly clear, and the paper would be suitable for publication with minor revisions. However, I would like the authors to clarify exactly what is new in this paper. It sounds like most of the methodology was already developed – is any of it new to this paper? I assume this is the first time the authors applied these methods to GF-4 data? Has similar work been published using GF-4 data?

Response:First of all, thank you very much for affirming this article. We have carefully read your questions and comments, and think that your suggestions are very helpful for us to revise and improve this article. According to your suggestions, we have made point-to-point modifications in the article. And the modified parts of the manuscript are marked with colors. We hope that our reply and revision will meet your requirements. We believe that we have unquestionably improved the quality of this paper with the amendments. Sincere thanks again!

Next, I will answer your question in the first paragraph of the comments. Since the visible light resolution of the GF-4 satellite is only 50 meters, it was originally used for meteorological monitoring in China, such as typhoon prediction. There is no precedent for using its visible band images for maritime target detection. In the accidental observation of the influence sequence, we found that the shape and trajectory of moving targets such as ships appear in the visible grayscale image. Because it is based on dynamic image sequence observation, moving targets are easily detected by the human eye. Thus, we have the idea of using the method based on inter-frame information to detect the target. From the point of innovation of detection methods, we used geographic radiographic transformation and pixel coordinate transformation as a bridge, ViBe target detection and CSRT target tracker are connected together, thus forming a complete system for GF-4 image target detection and tracking. For the ViBe method, we compared it with several background extraction methods through experiments. By comparison, the ViBe algorithm has a better effect and is very suitable for GF-4 images. Then we did not use its original default parameter settings. Instead, through experimental comparison, new parameter settings were obtained for the unique characteristics of the GF-4 image. This is mentioned in articles L208-L219.

Point 1:“Particularity” (Line 2, 5, 88, 388) This sounds strange to a native English speaker. I recommend changing it to something like “unique characteristics”. But there is another issue as well. The paper makes it sound like there is something wrong with the camera on GF-4. It sounds like the “particularity” has to do with the imagery. However, I think what the authors intend to highlight is that detecting ships with 50 m resolution imagery is a difficult task – so it is the use of this imagery for this application that gives this problem its unique characteristics (more than the imagery itself).

Response 1: Thank you very much for your comments. Based on your suggestions, we have replaced "particularity" with "unique characteristics". In addition, we very much agree with your opinion. The original intention of the paper should be that it is very difficult to detect the target in the low-resolution remote sensing image, so we use the characteristics of the GF-4 image to complete the target detection according to the method of information difference between frames. In order to let readers better understand the article, we have made changes in L92 to L105.

Point 2:Line 14, “aerospace” I recommend you either delete this word or replace it with “spaceborne”.

Response 2: Thank you for your suggestion, we strongly agree. It is indeed a problem of our English expression. We have replaced "aerospace" with "spaceborne" in L14 in accordance with your suggestion.

Point 3:Line 15-16, “How to make the computer” I get what you are trying to say, but that is awkward. Recommend changing it to something like “Development of automated algorithms to” or “Automating the process of”.

Response 3: Thank you for your suggestion. Based on your suggestions, we have revised the sentence of L15-17 in the text. Modify the original sentence to "The development of automated algorithms, which can accurately and timely extract useful information from massive remote sensing images, is a key issue in the application of remote sensing technology."

Point 4:Line 28, “According to the different sensors” This is the wrong usage of “according to”, which you could replace with “Given”.

Response 4: Thank you for your suggestion, we strongly agree. It is indeed a problem of our English expression. We have replaced “according to” with “Given” in the text at L28 according to your suggestion.

Point 5:Line 38, Shouldn’t “Evaluation” be “evaluation”?

Response 5: I am very sorry for our writing error, thank you for your correction. We have made changes to L38 in the text.

Point 6:Line 50, “proposed Spatial reflectivity distribution to search” was confusing, recommend:

“proposed, Spatial reflectivity distribution, to search”

Response 6: Thank you for your suggestion, we have made changes in the text L50 according to your suggestion.

Point 7:Line 75, results are Section 3, not 4

Response 7: We are very sorry for our writing error. We've rearranged the chapter numbers.

Point 8:Line 88, “No. 4” What does that mean? Is it the fourth image channel?

Response 8: Thank you for your correction. Here is our mistake, we have changed the original sentence of L93 to "However, due to the unique characteristics of the GF-4 satellite remote sensing image, it has brought great challenges to target detection.

Point 9:Line 99, “takes about 60” What are the units?

Response 9: "60" here means "60 photos". It is our mistake. It means "it takes about 60 photos to complete the coverage of the 10 million square kilometers of the Western Pacific Ocean". According to the content needs, we deleted this part.

Point 10:“geographic radiation transformation” (Line 112, 221, 227, 362) I really don’t know what you mean by this.

Response 10: Here is our expression error. The word should be "geographic affine transformation". Affine transformation is a kind of spatial rectangular coordinate transformation. It is a linear transformation between two-dimensional coordinates and two-dimensional coordinates. We have made modifications in these places in L112,258,261,378.

Point 11:Line 116, “geometric correction” If this is the language you prefer to use, that is fine, but I don’t think “correction” is really the right idea. There is nothing wrong with the images that needs to be corrected, they simply need to be geolocated. Factors such as sensor pointing aren’t really “errors”, but are parameters used to geolocate pixels in an image.

Response 11: Thank you for your comments and suggestions. Sorry, here is our expression error. The word should be "Geolocation correction". During the imaging process of the GF-4 satellite, due to various errors of the sensor, the geographic location information carried by the captured image is in error from the actual geographic location information. Therefore, the geographic location information correction method of remote sensing images is used to correct the geographic location of the image. We have corrected the error in the text. (L116, L303).

Point 12:Line 117 “3600M” Don’t you mean “36000 km”?

Response 12: Thanks for your comment. We have changed the "3600M" in L126 to "36000 km".

Point 13:Line 123 “sensor position and posture”. “Posture” is an odd word to use – I think you mean “pointing” or “orientation”.

Response 13: Thank you for your suggestion, what we want to express is indeed "the orientatoin of the sensor", so according to your suggestion, we changed "posture" to "orientation" in the text. (L123).

Point 14:Lines 128-130: something is screwed up with line numbering here, but in the line after 128, you have a “The” with no period before it.

Response 14: Thanks for your comment, after adjustment, we added the line number. And added period in front of "The".

Point 15:Line 164 “volatility” – I get the idea, but you really mean “variability”.

Response 15: Thank you for your comment. We agree with your opinion, we changed it to "random variability" in the text at L168.

Point 16:Figure 7 – Do you intend to have those Chinese characters in the figure?

Response 16: Here is our mistake. We have replaced Figure.7.

Point 17:Line 206-207, In these two sentences, it sounds like you are arguing both that parameter adjustment isn’t needed, and that it is. What exactly do you mean by “for more video sequences”? Do you mean that for this particular problem, the default values aren’t optimal? Or do you mean that different sequences need different parameters? Please clarify exactly what you are getting at.

Response 17: In [17], the author mentioned that ViBe parameters do not need to be adjusted. But in [22], for different types of video, the parameters are necessary to adjust. Just like our GF-4 image, we found it through experiments. Adjusting the parameters can make the detection effect better than the default parameters. "For more video sequences" here refers to more different types of videos, or videos of different types of targets. For this part of the expression, we have made changes in the text L211.

Point 18:Line 214, It would be nice to either give the values that you are using here, or reference where in the paper those values are.

Response 18: Thank you very much for your suggestions. In the experiment in Section 4, we clearly pointed out the parameter values of our experiment. The parameters of the ViBe algorithm are set as follows: the number of samples per pixel is 40, the search radius is 40, the #min index is 5, and the sub-sampling rate is 16. (L326- 3328)

Point 19:Figures 9 and 10: It would be really helpful if the panels had labels saying what they represent, e.g., “(a) Original”, … At least do this for the leftmost column.

Response 19: Thank you very much for your suggestions. We have added and modified the caption in Figure.9.

Point 20:Line 326: What is the “ghost effect”? This is the first time it is mentioned.

Response 20: The “ghost effect” means “False shadow effect”. False shadow effect is a common problem in detecting moving targets with background difference methods. When modeling the background initialization, the moving targets may be in the background, and they will produce false shadow after moving. In another case, when the moving targets in the scene change from motion to stillness, and then start moving, False shadow will also be generated. Other ghost-like situations are objects left in the background or moving objects that stop moving. To allow readers to better understand, we replace the ghost effect with "false shadow effect" in the text and explain it.

Point 21:Table 1. Accuracy is fine, but it combines the effects of true detections and false detections. Could you please also give the probability of detection and false alarm rate?

Response 21: Thanks for your suggestions, we have modified Table1. Added the probability of detection and false alarm rate of each algorithm.

Point 22:Figure 10: Please say what the colors represent in the figure caption.

Response 22: Thanks for your comments, we have added the meaning of different colors to the caption of Figure.11.

Point 23:Line 367: Change spelling from “Traking” to “Tracking”

Response 23: Thank you for your correction. Here is our writing error, we have corrected it in the article L392.

Point 24:Line 386: “access period is short” What do you mean by “access period”? Is that the latency in how long it takes between when the image is taken and when it is available for processing? Is it the length of time that the camera captures the scene?

Response 24: Thank you for your comments. Because the satellite changes the shooting angle, the observation area also changes. The access period of shooting satellites refers to the time interval from the first shooting to area A to the next shooting to area A.

Reviewer 3 Report

The authors have proposed an interesting framework on tracking moving targets in satellite images. Few suggestions:

- Can you provide more details about Gaofen-4 (GF-4)? Are the images from Gaofen-4 (GF-4) used in this manuscript publicly available? If yes, can you provide the download details.
- More details on related literature needs to be included. For example, refer to these works on satellite-based sensing papers:
(b) https://www.spiedigitallibrary.org/journals/Journal-of-Applied-Remote-Sensing/volume-13/issue-2/026511/Ship-detection-and-tracking-method-for-satellite-video-based-on/10.1117/1.JRS.13.026511.full
- The authors need to provide more details about Mass speckle removal algorithm. Is the mean filtering the only methodology used in this process?
- Figure 5 (b): what does the three axis indicate? Please mention.
- Why is Introduction marked as Section 0?
- What is the motivation of using VIBE algorithm for background extraction?
- In the update strategy (of Figure 7), how did you choose the value of #min?

Author Response

Response to the Reviewer 3 Comments:

The authors have proposed an interesting framework on tracking moving targets in satellite images. Few suggestions:

Point 1:- Can you provide more details about Gaofen-4 (GF-4)? Are the images from Gaofen-4 (GF-4) used in this manuscript publicly available? If yes, can you provide the download details.

Response 1: Thank you very much for your question. The images used in this manuscript were provided by the other party when we collaborated with relevant departments on the project. I am very sorry that we do not have permission to disclose these images. We only focus on technical research. However, I have provided several web sites that can obtain some GF-4 remote sensing images.

http://rsapp.nsmc.org.cn/webservice/?type=gf

http://www.rscloudmart.com/xuetang/xxg/detail/gf4#yangli

Point 2:- More details on related literature needs to be included. For example, refer to these works on satellite-based sensing papers:

(b)https://www.spiedigitallibrary.org/journals/Journal-of-Applied-Remote-Sensing/volume-13/issue-2/026511/Ship-detection-and-tracking-method-for-satellite-video-based-on/10.1117/1.JRS.13.026511.full

Response 2: Thank you very much for your comments. We have made changes in the text according to your requirements, and these two documents are cited. (L32,L55-61)。

Point 3:- The authors need to provide more details about Mass speckle removal algorithm. Is the mean filtering the only methodology used in this process?

Response 3: Thank you for your question. Yes, in the preprocessing part, we only use mean filtering. The "clouds" of blocks are smoothed and removed. In future research work, we will further explore the GF-4 satellite's cloud, fog and noise removal.

Point 4:- Figure 5 (b): what does the three axis indicate? Please mention.

Response 4: Thanks for your suggestions, we have modified and added in Figure.5 what the three axes represent. The x and y axes represent the pixel coordinates of the image, and the z axis represents the gray value of the image.

Point 5:- Why is Introduction marked as Section 0?

Response 5: Thank you for your question. I typeset this article according to the MDPI template. The first part of the official template is to send section.0 to start. So I followed this rule. Now, we have rearranged the chapter numbers. The introduction begins with Section 1.

Point 6:- What is the motivation of using VIBE algorithm for background extraction?

Response 6: Thank you for your question. Since the visible light resolution of the GF-4 satellite is only 50 meters, it was originally used for meteorological monitoring in China, such as typhoon prediction. There is no precedent for using its visible band images for maritime target detection. In the accidental observation of the influence sequence, we found that the shape and trajectory of moving targets such as ships appear in the visible grayscale image. Because it is based on dynamic image sequence observation, moving targets are easily detected by the human eye. Thus, we have the idea of using the method based on inter-frame information to detect the target. For the ViBe method, we compared it with several background extraction methods through experiments. By comparison, the ViBe algorithm has a better effect and is very suitable for GF-4 images.

Point 7:- In the update strategy (of Figure 7), how did you choose the value of #min?

Response 7: In [17], the author of ViBe pointed out in the article that the value of #min defaults to 2. In our actual experiment, we set a different #min value to find the #min suitable for GF-4 images is 5. Under such settings, the ViBe algorithm has a better detection effect on GF-5 images.

Round 2

Reviewer 3 Report

The authors have successfully addressed the comments. Proofread the manuscript one more time before the final submission.